# Efficient Isolation and Expansion of Limbal Melanocytes for Tissue Engineering

**DOI:** 10.3390/ijms24097827

**Published:** 2023-04-25

**Authors:** Naresh Polisetti, Thomas Reinhard, Günther Schlunck

**Affiliations:** Eye Center, Medical Center, Faculty of Medicine, University of Freiburg, 79106 Freiburg, Germany

**Keywords:** limbal melanocytes, limbal niche cells, limbal stem cells, isolation, expansion, melanin-producing cells, flow sorting

## Abstract

Limbal melanocytes (LMs) are found in the corneoscleral limbus basal epithelial layer and interact with neighboring limbal epithelial progenitor cells. The difficulty of isolating and cultivating LMs is due to the small fraction of LMs in the overall limbal population and the frequent contamination of primary cultures by other cell types. This has limited the research on freshly isolated LMs and the investigation of their biological significance in the maintenance of the limbal stem cell niche. Here, we describe an optimized protocol for the efficient isolation and expansion of LMs from cadaveric corneal limbal tissue using CD90 and CD117 as selective markers in fluorescence-activated cell sorting to obtain a pure population of LMs (CD90^−^ CD117^+^) with self-renewal capacity and sustained melanin production. The isolation of pure LMs from a single preparation enables direct transcriptomic and proteomic analyses, as well as functional studies on freshly isolated LMs, which can be considered the proper counterparts of LMs in vivo and have potential applications in tissue engineering.

## 1. Introduction

Limbal epithelial stem/progenitor cells (LEPCs), present in the basal layers of limbal epithelium, are responsible for the homeostasis of corneal epithelium. The self-renewal, quiescence, and differentiation status of LEPCs are regulated by the limbal niche, a specialized microenvironment proposed to be composed of a distinct extracellular matrix (ECM), limbal vasculature, and neighboring limbal niche cells (LNCs) [1,2]. The ECM composition affects LEPC fate through adhesion receptors and physical interactions, whereas surrounding LNCs provide diverse molecular signals as cues for LEPC maintenance and differentiation [3,4]. Limbal melanocytes (LMs), one of the major LNCs, are located in the basal layer of limbal epithelium and are associated with about 10 surrounding LEPCs, which form clusters comparable to melanin units in the skin [5]. Melanocytes are thought to protect neighboring LEPCs against UV damage by transferring melanin-containing melanosomes to the surrounding LEPCs. The degree of pigmentation was found to correspond with the state of LEPC differentiation, with the most immature progenitor cells showing the most pigmentation [6]. LMs have been also shown to support corneal epithelial regeneration during wound healing and the maintenance of the LEPC phenotype, both in vitro and in vivo [7,8,9,10,11]. In addition, LMs were shown to have potent immunomodulatory, anti-inflammatory, and anti-angiogenic properties, making them potentially attractive for clinical use [9]. Thus, the co-cultivation of LEPCs with LMs might represent an improved strategy to generate cell transplants for patients suffering from limbal stem cell deficiency [12,13]. However, the cultivation of melanocytes has proven difficult due to the low percentage of melanocytes in the total limbal population and their low mitotic activity compared with fibroblasts and epithelial cells [7,10].

Few reports have been published on obtaining pure melanocyte cultures from limbal tissue using G418 (geneticin), a cytotoxic agent that affects rapidly growing cells [7,10]. Despite the fact that this cytotoxic procedure provides pure melanocyte populations, subtle toxic side effects on the phenotypic and functional features of the surviving cells are still possible [14]. To circumvent these constraints, Hayashi and coworkers isolated LMs based on N-cadherin expression using fluorescence-activated cell sorting (FACS) [15]. This method, however, is limited by the fact that only a small fraction of LMs express N-cadherin [15]. We have developed a protocol for isolating melanocytes based on CD117 (c-Kit) expression [16]. However, this protocol also requires at least one passage in culture to eliminate contaminating stromal cells and achieve a pure LM population [16]. Thus, research on freshly isolated LMs has been hampered by the lack of a good protocol for isolating LMs.

The CD90 molecule, also known as Thy-1, is a cell surface glycoprotein that is commonly expressed on stromal cells [11,17,18]. CD117, also known as c-Kit, is a receptor tyrosine kinase that is expressed in a variety of cell types, including melanocytes, and has been used to isolate LMs [11,16,19] and epidermal melanocytes [20]. Recently, we reported an efficient approach for isolating LMs from organ-cultured corneoscleral samples using fluorescence-activated cell sorting (FACS) with CD90 and CD117 as selective markers [11]. The core of this innovative method is the simultaneous extraction of stromal cells from the total limbal cell population using CD90, which minimizes the risk of stromal contamination. We proposed that flow sorting yielded a pure population of LMs (CD90^−^ CD117^+^; 1.2 ± 0.3% cells) with self-renewal capacity and sustained melanin production [11]. Despite the fact that our procedure yielded fewer melanocytes than other methods, such as magnetic-activated cell sorting using CD117 (1 to 7.0%) [19] and FACS employing only the anti-CD117 antibody (1 to 7.0%) [16], these methods require an additional purification step or at least one passage in culture to obtain pure melanocytes. In the current protocol article, we describe a step-by-step protocol to isolate and propagate human LMs for potential use in tissue engineering.

## 2. Experimental Design

Limbal melanocytes were isolated from organ-cultured corneoscleral tissues through a combination of enzymatic digestion and cell sorting. The limbal segments were digested with collagenase A to obtain limbal clusters, which consisted of epithelial–mesenchymal melanocytes and single cells. The obtained limbal clusters were dissociated into single cells using trypsin and ethylenediaminetetraacetic acid (EDTA). The obtained single-cell suspension was stained with anti-CD90 and anti-CD117 antibodies, and FACS sorting was performed. The CD90^−^ CD117^+^ (limbal melanocytes) cells were sorted and cultured. An overview of the stages of the protocol is presented in Figure 1.

### 2.1. Materials and Reagents

#### 2.1.1. Glassware and Plasticware

The following glassware and plasticware were used in this protocol. However, you may test any other available commercial source of the same requirements.

➢Micropipette tips (0.5–20 µL, 100–200 µL, 1000 µL) (Greiner Bio-One, Frickenhausen, Germany);➢A 12-well plate (Corning, Costar^®^, Kaiserslautern, Germany, catalog number: 3513);➢A 60 mm cell culture dish (Corning, Falcon^®^, catalog number: 353004);➢A 100 mm cell culture dish (Corning, Falcon^®^, catalog number: 353003);➢Syringe filter 0.2 μm (VWR, Radnor, PA, USA, catalog number: 28145-501);➢Disposable scalpel blades, No. 10 (pfm Medical ag, Feather^®^, catalog number: 201000010);➢Serological pipettes (5 mL, 10 mL) (Corning, Stripette™);➢Conical tubes (15 mL) (Greiner Bio-One, catalog number: 188271);➢Conical tubes (50 mL) (Greiner Bio-One, catalog number: 227261);➢T75 flasks (Corning, catalog number: CLS430641);➢Reversible cell strainers (Stem Cell Technologies, Köln, Germany, catalog number: 27215);➢Cell filter 20 µm (Cell Trics™, Sysmex Partec GmbH, Norderstedt, Germany, catalog number: 04-004-2325);➢FACS tubes (5 mL polystyrene round-bottom tube, Falcon, catalog number: 352058).

#### 2.1.2. Antibodies

Cell-specific antibodies and their corresponding isotype controls were required for cell sorting and characterization of cells. Anti-CD117 antibody was used as a selective marker for limbal melanocytes, whereas anti-CD90 was used to identify limbal mesenchymal stromal cells. For characterization, epithelial marker pan-cytokeratin; stromal marker vimentin; and melanocyte markers Melan-A, SRY-box-10 (Sox10), human melanoma black (HMB)-45, and tyrosinase-related protein (TRP)1 were used.

➢PE mouse anti-human CD117 (BD Biosciences, Heidelberg, Germany, catalog number: 561682);➢APC mouse anti-human CD90 (BD Biosciences, catalog number: 55986);➢Mouse IgG2a, k isotype-APC (eBioscience, Thermo Scientific, Dreieich, Germany, catalog number: 17-4724-81);➢Mouse IgG2a, k isotype-PE (Biolegend, Amsterdam, The Netherlands, catalog number: 400212);➢Mouse anti-human cytokeratin pan AE1/AE3 (DAKO, Hamburg, Germany, catalog number: M3515);➢Rat anti-human vimentin (R&D Systems, Wiesbaden, Germany, catalog number: MAB2105);➢Rabbit anti-human Melan-A (Abcam, Cambridge, UK, catalog number: ab210546);➢Rabbit anti-human Sox10 (Abcam, catalog number: ab155279);➢Mouse anti-human HMB-45 (Abcam, catalog number: ab787);➢Mouse anti-human TRP1 (Abcam, catalog number: ab235447).

You may test any other manufacturers’ antibodies of the same type.

#### 2.1.3. Culture Media, Reagents, and Buffers

➢For washing tissues and cells, Dulbecco’s Phosphate Buffered Saline (DPBS) (no calcium, no magnesium) (Thermo Fisher Scientific, Dreieich, Germany, Gibco™, catalog number: 14190094) was used;➢Dulbecco’s Modified Eagle Medium (DMEM) high glucose (Thermo Fisher Scientific, Gibco™, catalog number: 11960044) and fetal bovine serum (FBS) (Thermo Scientific, Gibco™, catalog number: 10082147) were used to inhibit trypsin action;➢To propagate LMs, we recommend using CnT-40 melanocyte proliferation medium (CELLnTEC, Bern, Switzerland);➢For successful expansion of LMs, we used iMATRIX 511 (Nippi, Tokyo, Japan, catalog number: 892012) to coat the cultureware;➢Penicillin–streptomycin 100× (Sigma-Aldrich, Taufkirchen, Germany, catalog number: P4333); ➢EDTA 0.5 M (Invitrogen, Karlsruhe, Germany, catalog number: AM9260G) was used in FACS buffer to reduce cell clump formation;➢As a disinfectant, 70% isopropyl alcohol was used;➢4′,6-diamidino-2-phenylindole (DAPI) (Sigma-Aldrich, catalog number: MBD0015) was used during FACS to differentiate live and dead cells. You may experiment with any viability indicator that works well with the conjugated antibodies used in sorting.

#### 2.1.4. Cell and Tissue Dissociation Reagents

➢Collagenase A (Sigma-Aldrich, Roche Diagnostics, Basel, Switzerland, catalog number: 10103578001) was used to digest the limbal segments;➢Trypsin–EDTA 0.25% (Thermo Fisher Scientific, Gibco^®^, catalog number: 25200056) was used to dissociate limbal clusters into single cells as well as passaging of limbal melanocytes.

### 2.2. Equipment Required

➢Pipettes: micropipettes of all sizes (P20, P200, P1000; Eppendorf Research plus) and pipette aids (BRAND, accu-jet^®^ pro) were required;➢Centrifuge: the centrifuge must be capable of reaching speeds of at least 400× *g* with rotors to accommodate 15 mL and 50 mL centrifuge tubes and FACS tubes;➢Cell counter: In this study, we used hemocytometer (MARIENFELD, Lauda-Königshofen, Germany, catalog number: 0640130). You may use any kind of cell counter to count the cell numbers;➢Surgical instruments: to dissect the tissues, forceps (Dumont, Hausen ob Verena, Germany, 5-Dumoxel^®^-H) were required;➢Aseptic environment: A biosafety cabinet (Thermo Scientific™, Type S2020 1.2) was required to create an aseptic environment for cell isolation and culturing. All the steps need to be performed under a biosafety cabinet;➢CO_2_ incubator: for optimal cell culturing, CO_2_ incubator (Thermo Scientific™, Heracell™ 240i) was required to maintain 37 °C and 5% CO_2_;➢Microscope: to visualize cell phases, contrast-inverted microscope with a camera (ZEISS, Oberkochen, Germany, Objectives 4×, 10×, 20×) was required;➢Refrigerator and freezer: a refrigerator at 2–8 °C and freezer at −20 °C were required for storage of media and reagents;➢Flow sorter: In this study, we used FACS Aria II sorter (BD Biosciences) to sort the cells. You may use any other cell sorter to sort the cells using this protocol;➢Water bath: a water bath (GFL^®^, Karlsbad, Germany, catalog number: 1013) was required to warm the reagents.

### 2.3. Software Required

➢CapturePro 2.10.0.1 (JENOPTIC Optical systems GmbH; Villingen-Schwenningen, Germany);➢FACSDiva v8.0.1 software (BD Pharmingen, BD Biosciences, Heidelberg, Germany);➢FlowJo v10.2 software (Tree Star, Ashland, OR, USA).

### 2.4. Preparation of Solutions

Collagenase solution: to dissect corneolimbal tissue, collagenase solution was required. Collagenase solution (2 mg/mL) was prepared and stored at −20 °C. To prepare collagenase solution, dissolve 500 mg of collagenase A in 220 mL of DMEM high glucose containing 25 mL of fetal calf serum and 5 mL of penicillin–streptomycin 100×. Mix well by inverting until it dissolves completely. Filter the solution using 0.2 µm filter, aliquot as desired (5 or 10 mL aliquots), and store at −20 °C. Avoid repeated freeze–thawing.

CnT-40 medium: thaw the CNT-40 medium at 2–8 °C overnight. Then, add 5 mL of penicillin–streptomycin 100× to 500 mL of CnT-40 medium. Mix by inverting, and prepare aliquots if needed. Store at 2–8 °C, and use within one month.

FACS buffer: to prepare FACS buffer, add 1 mL of FBS and 25 µL of EDTA (0.5 mM EDTA) to 24 mL of DPBS. Mix well by inverting. Always prepare a fresh solution.

Laminin-coated plates: coat the culture plates with iMatrix-511 according to the manufacturer’s instructions. We used 0.5 µg/cm^2^ to coat the cultureware. For coating of the T75 culture flask, dilute 75 µL of iMatrix-511 in 8 mL of 1× DPBS. Coat the T75 flask with 8 mL of dilute iMatrix-511 (0.5 µg/cm^2^) solution for 1 h at 37 °C, 5% CO_2_ in the incubator, or at 4 °C overnight. After incubation, remove the iMatrix-511 solution, and add 10 mL of CnT-40 medium until use.

## 3. Procedure

The dissection of the limbus and preparation of limbal clusters is similar to the previously published procedure [21,22].

**A.** Tissue1.Organ-cultured corneoscleral tissues (age of 64.5 ± 12.2 years; culture duration 23.0 ± 3.5 days; postmortem time of 32.66 ± 15.8 h) were obtained from the cornea bank with appropriate research consent and ethical approval [11]. Tissues that are not suitable for transplantation due to low corneal endothelial cell density (<2200 cells/mm^2^) or the presence of neurological disorders or malignancies in the donors can be used. Donor cornea remnants after corneal endothelial transplant preparation are also a valuable source if appropriate research consent has been obtained.**B.** Dissection of limbus2.Place the organ-cultured corneoscleral tissue in a 60 mm culture dish, and wash twice with DPBS. Using a scalpel blade and forceps, scrape the posterior side (endothelial side) of the tissue to remove unwanted tissue, such as trabecular meshwork and iris/ciliary tissue;3.Place the tissue in a new 60 mm culture dish containing 1× DPBS, and cut the tissue into four equal quarters (Figure 2A) of corneoscleral segments (Figure 2B) using a scalpel blade and forceps;Note: to obtain adequate LMs for downstream applications, use 4–6 corneoscleral tissues.4.Make incisions of corneoscleral segments at 1 mm before and beyond the anatomical limbus to obtain limbal segments (Figure 2C).**C.** Preparation of limbal cell suspension5.Place the limbal segments in a 60 mm dish with 5 mL of collagenase A (2 mg/mL), and cut them into smaller pieces (each limbal segment into 2–3 pieces) with a scalpel blade. Incubate overnight at 37 °C with 5% CO_2_ to digest the stromal collagen and obtain limbal cell clusters;6.After overnight incubation (~16–18 h), triturate the suspension with an up-and-down motion using a 1 mL pipette (P1000), and examine for the presence of cell clusters and single cells under the microscope (Figure 2D);Note: If incomplete digestion of limbal segments occurs after overnight incubation trituration, re-incubate for an additional 2 h in the same solution at 37 °C with 5% to ensure complete digestion. In contrast, prolonged digestion of tissue (more than 20 h) may have a negative impact on cell viability and quality.7.To separate limbal cell clusters from single cells, filter the tissue digest using 20 µm cell filters that allow single cells to flow through while retaining the clusters;8.To remove any remaining single cells, wash the filters twice with DPBS;9.Place the strainer in the opposite direction on a 60 mm dish. Add 5 mL of 0.25% trypsin–EDTA to flush clusters (Figure 2E) into a Petri dish, and incubate at 37 °C (incubator) for 10–15 min to dissociate the clusters into single cells;Note: In place of 20 µm cell filters, 37 µm reversible cell strainers can be used. Do not allow the filter to dry as this might cause cell death. The filtered single-cell suspension can be discarded or used for other purposes, such as obtaining limbal fibroblasts.10.After incubation, triturate the cell suspension with an up-and-down motion using a 1 mL pipette. Observe the cell suspension under the microscope to ensure the complete dissociation of clusters into single cells (Figure 2F). Add 5 mL of pre-warmed DMEM (37 °C in water bath) containing 10% FBS to inhibit trypsin digestion;Note: If clusters are not completely dissociated after 15 min of incubation and trituration, re-incubate for an additional 5 min in the same solution at 37 °C with 5% CO_2_ to achieve complete dissociation. Prolonged incubation of clusters in trypsin–EDTA might adversely affect cell viability and quality.11.Filter the cell suspension using a 40 µm cell strainer into a 15 mL Falcon tube to remove any cell clumps. Wash the cell strainer with DPBS twice;12.Centrifuge at 200× *g* for 5 min at room temperature;13.After centrifugation, remove the supernatant carefully, and resuspend the cell pellet in 200 µL of ice-cold FACS buffer (see Section 2.4) by pipetting up and down using a P200 pipette.**D.** Fluorescence-activated cell sorting (FACS)14.Transfer the cell suspension to FACS tubes (100 µL/tube). Add a mouse APC-conjugated anti-human CD90 antibody (5 µL/106 cells) and PE-conjugated anti-human CD117 antibody (5 µL/106 cells) to one tube and an IgG2a-Isotype APC and an IgG2a-Isotype PE to another tube at 4 °C;Note: The cell density should be <1 × 10^7^ cells/mL. If the cell number is high, adjust the volumes and concentration of antibodies accordingly.15.Incubate the cells on ice for 45 min, agitating every 15 min16.After incubation, add 1 mL of FACS buffer to each FACS tube, and centrifuge for 5 min at 400× *g* at 4 °C. Repeat the washing process twice;17.After washing, add 500 µL of FACS buffer containing DAPI (1:5000) to the cell pellet to exclude dead cells, and proceed to flow sorting using a FACS Aria II sorter [11];18.Set up the FACS Aria II machine according to manufacturer’s instructions;19.Analyze the limbal cell suspensions. Set the gate on forward scatter (FSC-A) and side scatter (SSC-A) to select cells of interest based on size and granularity (Figure 3A). To remove doublets or clumps and enrich single cells, employ side scatter area vs. width (Figure 3B), followed by dead cell exclusion based on DAPI (Figure 3C);20.Then, depending on the isotype controls, adjust the gates to select CD90^−^ CD117^+^ (Figure 3D);Note: a positive region for each antibody staining needs to be defined, and this is defined as a region that contains almost no cells (e.g., <0.3%) when stained with the corresponding isotype control antibody.21.Sort the CD90^−^ CD117^+^ cells into 2 mL of FACS buffer in FACS tubes.Note: the CD90^−^ CD117^−^ populations mainly contain LEPCs and can be used to enrich LEPCs using cell type-specific media, whereas the CD90^+^ population containing LMSCs can be used for the expansion of LMSCs in cell type-specific media [11].**E.** Expansion of limbal melanocytes22.Seed all the sorted CD90^−^ CD117^+^ LMs into a well of a 12-well plate coated with i-Matrix-511 (see Section 2.4, laminin-coated plates);Note: the number of CD90^−^ CD117^+^ cells per limbus varies from sample to sample.23.Cultivate the LMs at 37 °C with 5% CO_2_ in CnT-40 medium (2 mL/well) to expand LMs. Change media every 2 days. Visualize the morphology of LMs using phase-contrast microscopy.**F.** Sub-cultivation of limbal melanocytes24.Remove the media from the culture vessel at 70 to 80% confluency;25.Wash the cells using DPBS, and add 1 mL (per well 12-well plate) of trypsin–EDTA (0.25%; pre-warmed at 37 °C in a water bath). Incubate at 37 °C with 5% CO_2_ for 5 min;26.After incubation, add 1 mL of DMEM containing 10% FBS to inhibit trypsin action, and mix well;27.Transfer the cell suspension to a 15 mL tube, and centrifuge at 200× *g* for 5 min. Remove the supernatant carefully, resuspend the cell pellet in CnT-40 medium and count the total cell number using a hemocytometer;Note: after 10 to 15 days of culture (roughly 70 to 80% confluency), the number of cells obtained from the well of a 12-well culture plate ranges from 40 to 60,000 cells.28.Seed the cells in T25 flask (2 × 10^3^
cells/cm^2^, i.e., 50,000 cells/T25 flask) coated with i-Matrix-511 in CnT-40 medium, and change the media every other day;Note: Over-confluence (more than 80%) and prolonged trypsin digestion (more than 5 min) adversely affect cell viability and the quality of cells during sub-culturing. Always passage cells at 70 to 80% confluence. Avoid prolonged incubations in trypsin. We recommend the seeding density of 2 × 10^3^ cells/cm^2^ for expansion of LMs. If the obtained cell number is lower than 50,000 cells, we still recommend seeding all the cells in T25 flask.29.After 10 to 14 days, repeat the steps from 24 to 27. Use the cells for the application of choice or sub-culturing. For subculturing, seed all the cells in T75 flask coated with i-Matrix-511 (see Section 2.4) in CnT-40 medium, and change the media every other day. Incubate at 37 °C with 5% CO_2_ for 5 min;30.Passage the cells (2 × 10^3^ cells/cm^2^) until the desired number of LMs is achieved for application of choice. Melanocytes take 10 to 14 days to reach confluence (with seeding density of 2 × 10^3^ cells/cm^2^). The culturing and passaging of LMs can be carried out for 24 months without any significant changes in cell proliferation or behavior;31.To evaluate the LM characteristics, their phenotypic profile, colony-forming efficiency, growth characteristics, and functional characteristics are tested (Figure 4). Please refer to the published article for the detailed protocols [11,23].

## 4. Expected Results

The conditions provided in this protocol were optimized to isolate and expand LMs. A detailed analysis of the isolation and expansion of the LMs can be found in a study by Polisetti et al. (2022) [11]. Briefly, the collagenase A digestion of limbal segments generated single cells as well as clusters (Figure 2D) containing niche populations of LEPCs, LMSCs, and LMs. Trypsin/EDTA treatment dissociated cell clusters into single cells (Figure 2F). The number of viable single cells retrieved from clusters ranged from 40,000 to 75,000/corneoscleral tissues. These single cells were stained for CD117 and CD90 and subjected to flow sorting. To identify cells of interest based on size and granularity, the limbal cell suspensions were gated on forward scatter (FSC-A) and side scatter (SSC-A) (Figure 3A). The side scatter area vs. height was used to enrich single cells to eliminate doublets or clumps (Figure 3B), followed by dead cell exclusion using DAPI (Figure 3C). Then, based on the isotype controls, gates were set to pick CD90^−^ CD117^+^ cells (Figure 3D). Limbal cluster-derived cell suspensions from donor corneoscleral samples provided a yield of 1.2 ± 0.3% of CD90^−^ CD117^+^ cells (Figure 3E). The number of CD90^−^ CD117^+^ (200–1050) per limbus varied between samples. The cultured CD90^−^ CD117^+^ cells appeared large and flattened, with smooth bodies and multiple dendrites under phase-contrast microscopy (Figure 4A) and stain PCK^−^ Vimentin^+^/Melan-A^+^ (Figure 4B), which are characteristics of melanocytes.

The specific marker expression, self-renewal capacity, and functional studies were investigated (the test procedures can be found in our previously published article [11]). In all cultured CD90^−^ CD117^+^ cells, immunostaining confirmed the expression of human melanoma black-45 (HMB-45), SRY-box transcription factor 10 (Sox10), and tyrosinase-related protein 1 (TRP1) (green) (Figure 4C). The self-renewal potential of LMs was evaluated by seeding cells at a low density (20 cells/cm^2^) (Figure 4D). The colony-forming efficiency was 81.0 ± 34.0%. Melanin production is a primary function of melanocytes, and it plays an important role in many biological processes, particularly protection against the harmful impacts of UV radiation. Melanin production was assessed by culturing LMs in the presence (treated) or absence (untreated) of L-3,4-dihydroxyphenylalanine (L-DOPA). The wells (12-well plate) of LMs (P1) cultured in the presence of 1 mM of L-DOPA for 24 h showed light brown coloring of the culture medium in the treated wells, as was observed macroscopically (Figure 4E). These findings indicate that enriched CD90^−^ CD117^+^ LMs are functional in producing and secreting melanin [11].

During the isolation of LMs from donor limbal tissue, reagent preparations, or the culture of LMs, some problems described in Table 1 may be encountered.

In conclusion, we described a step-by-step protocol for obtaining pure and functional LMs from organ-cultured corneoscleral tissues in less than a day. The unpassaged LM populations obtained enable direct transcriptomic analysis, proteomic profiling, stem cell-based tissue engineering approaches, and functional studies to further explore melanocytes’ role in the limbal stem cell niche.

## Figures and Tables

**Figure 1 ijms-24-07827-f001:**
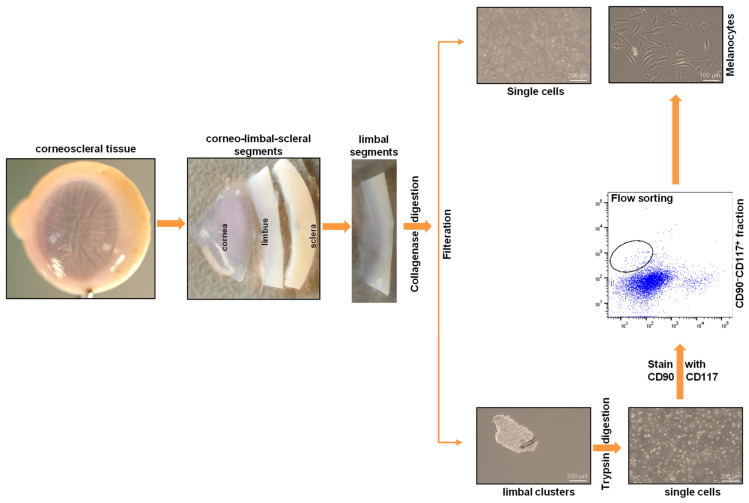
An overview of the stages of the limbal melanocyte isolation protocol: Dissect the corneoscleral tissue into limbal segments, followed by digestion of limbal segments using collagenase to produce limbal clusters and single cells. Dissociate the obtained limbal clusters into single cells using trypsin. Stain the single-cell suspension with anti-CD90 and anti-CD117 antibodies and perform flow sorting to obtain CD90^−^ CD117^+^ cells (limbal melanocytes).

**Figure 2 ijms-24-07827-f002:**
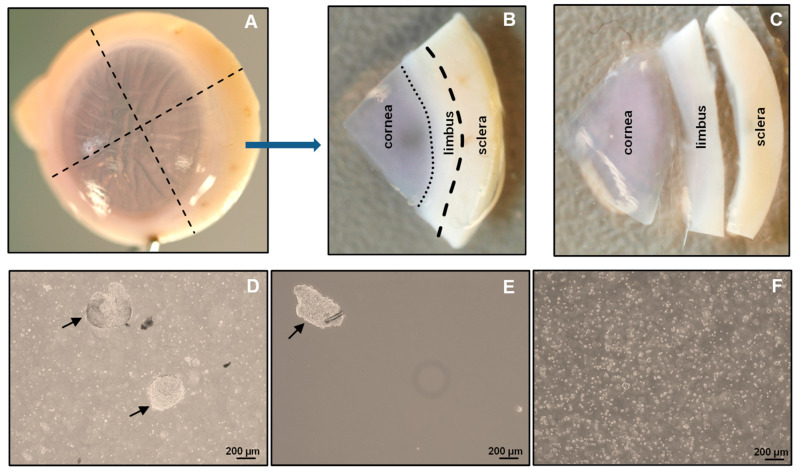
Isolation of limbal cluster cells: The corneoscleral tissue (**A**) was cut into four corneoscleral sectors (**B**), and each sector was trimmed 1 mm before and after the limbal region (**C**). (**D**) Different sizes of limbal clusters (arrows) and single cells formed after overnight incubation of limbal segments in collagenase solution. (**E**) Limbal clusters (arrow) separated from single cells after filtration. (**F**) Single-cell suspension of limbal cells after digestion of limbal clusters with trypsin–EDTA.

**Figure 3 ijms-24-07827-f003:**
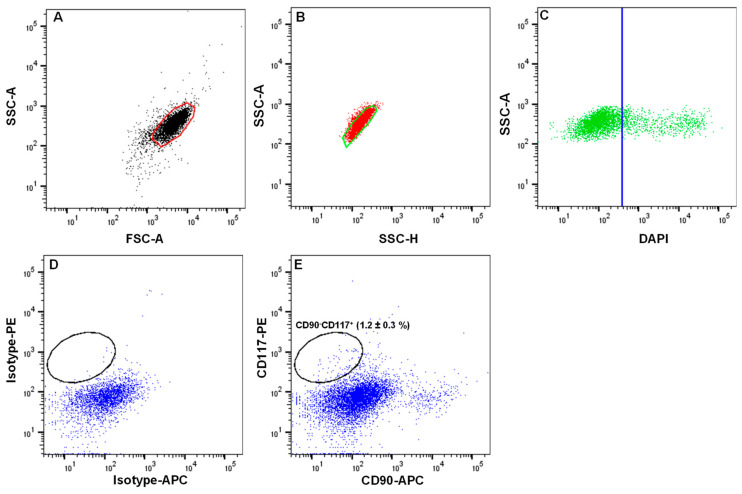
Fluorescence-activated cell sorting (FACS) images demonstrating the gating strategy used to isolate limbal melanocytes. (**A**) Forward scatter (FSC-A) vs. side scatter (SSC-A) graph showing the cells of interest selected on the basis of size and granularity. Side scatter area vs. height graph showing the selection of single cells by excluding doublets or clumps (**B**), followed by dead cell exclusion using 4′,6-diamidino-2-phenylindole (DAPI) (**C**). The isotype control graph shows the set of gates (**D**) used to select the CD90^−^ CD117^+^ cells (**E**). Percentages (%) of positive cells are expressed as the means ± SEM.

**Figure 4 ijms-24-07827-f004:**
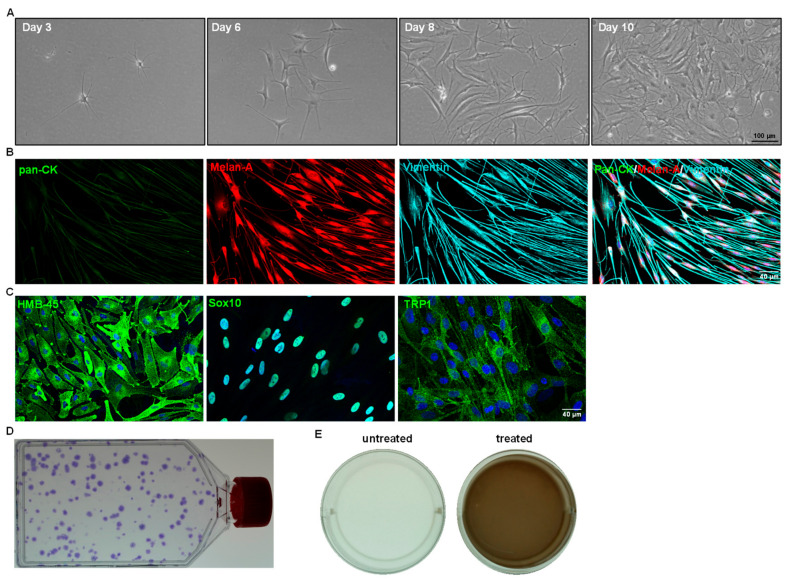
Cell culture characteristics of sorted limbal melanocytes. (**A**) Phase contrast images showing the CD90^−^ CD117^+^ cells after days 3, 6 (body rounded with multiple dendrites), 8, and 10 (flattened, smooth bodies with multiple dendrites) of seeding (×100 magnification). (**B**) Triple immunostaining analysis of cultured cells showing Melan-A (red) and vimentin (cyan) double-positive cells and negative for epithelial keratins (pan-CK, green). Nuclear counterstaining with 4′,6-diamidino-2-phenylindole (blue). (**C**) Immunocytochemical analysis of cultured CD90^−^ CD117^+^ LMs showing the expression of human melanoma black-45, sex-related HMG box 10 (Sox-10), tyrosinase-related protein 1 (TRP1) (green), and nuclear counterstaining with DAPI (blue). (**D**) T75 flask showing crystal violet-stained colonies of LMs. (**E**) Melanin production was assessed by culturing the LMs in presence (treated) or absence (untreated) of L-3,4-dihydroxyphenylalanine (L-DOPA). The wells of LMs in the presence or absence of 1 mM L-DOPA for 24 h showing light brown coloring of the culture medium in treated wells, as observed macroscopically.

**Table 1 ijms-24-07827-t001:** Contains possible explanations and solutions to aid researchers in troubleshooting.

Problem Encountered	Explanations	Solutions
Limbal tissue is not digested completely or at all	Collagenase A might have degraded	Prepare fresh collagenase Ensure that the stock solution and collagenase A powder are stored properly
	Incomplete digestion	Extend the incubation time but not over 20 h A higher concentration of collagenase (2%) can be used to digest limbal tissue
Limbal clusters are not or only partially digested	Trypsin–EDTA might have degraded	Use fresh trypsin–EDTA Ensure proper storage of stock solution
	Incomplete digestion	Prolong the incubation time but do not exceed 20 min
Yield of sorted LMs is low	Sorted LMs might have been lost in the collection tubes	Use low-absorbing tubes
	Quality of donor tissue	Obtain only healthy limbal tissues from younger donors (set a cutoff age of 70 y) if possible. The younger the donors, the higher the number of melanocytes
	A prolonged period of tissue digestion caused cell death	Do not exceed 20 h digestion in collagenase A
LMs did not attach/grow	Laminin-coated plates dried	Ensure the laminin plates do not dry while coating the plates Always use freshly coated plates. If prepared in advance, keep at 4 °C for no longer than one week
	Culture medium is not in optimal condition	Always use CnT-40 fresh medium. Do not use medium kept for more than a month at 4 °C
Stromal contamination	Gating strategy is not accurate	Set the gates according to the corresponding isotype controls (0.3%)
Microbial contamination	Donor corneal tissue is infected	If the cornea looks cloudy/hazy, do not process
	Contamination during cell processing	Ensure that surgical instruments are autoclaved, and all materials and reagents used are sterile Add antibiotics to the FACS buffer

## Data Availability

The datasets generated during and/or analyzed during the current study are available from the corresponding author on reasonable request.

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
