# Peer review of "Efficient Isolation and Expansion of Limbal Melanocytes for Tissue Engineering"

_ijms, 2023, doi:10.3390/ijms24097827_

Round 1

Reviewer 1 Report

IJMS-2326483.

This manuscript is purely methodological. Authors describe in detail the procedures followed for dissociation, isolation and cultivation of limbal melanocytes from corneo-limbal-scleral rings obtained from cadaveric tissue, and if carried out as described, it is possible to obtain limbal melanocytes after FACS selecting CD90- CD117+ cells. This manuscript refers to previously described methods (Li et al., 2022, Int J Mol Sci. 23(7):3756; Polisetti et al., 2022, Int J Mol Sci. 23(5):2750), in this case, with the difference centered in limbal melanocyte isolation and cultivation.

MAJOR ISSUE.

In view of the approach presented in this manuscript, there is no novelty regarding to the previous published work. Taking into account that cell yield per sample is too low, It should be necessary that authors include some data about the potential of the described cell culture conditions to carry out serial transfer of the melanocytes to obtain enough cells for other experiments or to attempt co-cultivation with limbal epithelial cells as proposed.

Reviewer 3 Report

Polisetti et al's aim in this paper was to present a streamlined approach for the isolation and expansion of limbal melanocytes (LM) that can be used in tissue engineering and other downstream applications. The authors have adapted this protocol from their earlier work and have provided a clear and concise explanation of the protocol and potential troubleshooting measures. However, some concerns have been raised to improve the manuscript's quality.

One significant concern is the novelty of the study and the protocol. Although the authors have described the experimental design and protocol in great detail, there are no new findings or information on this technique. It appears to be an adaptation of the LM isolation method previously published by the authors. As a result, this protocol should have been reported as a brief report.

In the last paragraph of the Introduction section, the authors mentioned that their proposed method yielded a pure population of LM (CD90−CD117+). The authors should state the percentage purity of LM obtained by their approach and compare it with other established methods.

Furthermore, the image quality of Figure 1 needs improvement.

Reviewer 4 Report

In the described protocol for isolating Limbal melanocytes (LM), CD90 and CD117 were used as selective markers for fluorescence-activated cell sorting (FACS). These markers were selected based on their expression pattern in LM and their ability to isolate pure LM populations with self-renewal capacity and sustained melanin production. The CD90 molecule, also known as Thy-1, is a cell surface glycoprotein that is commonly expressed on stromal cells and has been used as a selective marker for isolating LM. CD117, also known as c-Kit, is a receptor tyrosine kinase that is expressed on a variety of cell types, including melanocytes, and has been used to isolate LM from other cell types in the limbal niche. By using these markers in combination, the described protocol was able to isolate a pure population of LM that can be used for further studies on their biological significance in maintaining the limbal stem cell niche and potential applications in tissue engineering.

Manuscript is written in a very weird shape.

Comment 1:  text has different font size and I is not properly organized

Comment 2: figure legend are not aligned with figures  

Comment 3: figure 2 does not have scale bar in F

Comment 4: method sections are not written in a version that should be written.

Comment 5: Why it should have expected results? It should either have results or not.

Comment 6Figure 4 does not have scale bar. Section E looks like schematic photo? There is not a good explanation for this.

Comment 7: language used in this manuscript has so much “not certain” tone.

Comment 8: There is not conclusion section.

Reviewer 5 Report

The manuscript describes an efficient method for the isolation and expansion of limbal melanocytes derived from organ-cultured corneoscleral tissues. The procedures have been explained meticulously and results have been presented clearly. Below are my comments for the authors:

1.      Is there any evidence regarding the percentage of limbal melanocytes in the basal layer of human limbal epithelium? Is it different between the quarters (i.e., superior/inferior versus nasal/temporal)? And how much variable is it between different individuals?

2.      How robust are the markers that have been used to differentiate limbal melanocytes from other limbal cells? Please discuss the specificity of CD90- CD117+ for limbal melanocytes.

3.      Table 1, solutions: Would the authors recommend using Dispase II as an alternative to collagenase A for the digestion of the limbal epithelium?

4.      Please discuss the methods and results of recently published papers, like PMID: 33865984 and PMID: 35409129, and compare them with the methods and findings described in this manuscript.

Round 2

Reviewer 1 Report

Authors had made changes that clarify the doubts raised about this manuscript. As a methodological report is adequate.

Reviewer 3 Report

The manuscript attempted to tackle certain problems I had, but it did not offer any fresh insights. The authors did not satisfactorily address my concern, as they merely adopted the LM isolation method already published by the same group in IJMS. Consequently, this protocol seems to be a replication of their prior research, and it would have been appropriate to report it as a brief report.

Reviewer 4 Report

Revised version is in a good shape for publication. 

Reviewer 5 Report

The authors have responded to my comments satisfactorily. I have no further comments on the revised manuscript.